# Exploring a novel Feedback Mechanism for Convolutional Neural Networks

## Abstract

Convolutional neural networks (CNNs), which have achieved significant success in various visual tasks, are inspired by the architecture of the mammalian vision system. However, unlike CNNs, the visual cortex contains a substantial number of top-down or feedback connections. Inspired by this, recent research has investigated incorporating feedback mechanisms into CNNs. In this paper, we propose a novel feedback mechanism called 'Image Specific Feature Selection (ISFS)' that leverages feedback to utilize only a relevant subset of filters for the given image. The feedback weights are learned, and thus the network learns to select features/filters tailored to each image. The feedback improves performance both in terms of better accuracy and better confidence in classification. The selection of filters through the feedback is indeed image-specific and results in interesting behaviour of the network. The feedback signals produced for a given image, can be viewed as a useful low-dimensional approximation of the internal representation of the image. We demonstrate that we can effectively use the feedback signals to identify when a given image has adversarial noise.

**Keywords : CNN, Feedback in CNNs, Image Specific Feature Selection**

## 1 Introduction

Convolutional Neural Networks (CNNs) have been highly successful in a variety of image processing tasks. Many of the architectural features of CNNs, such as multiple layers of processing, hierarchy of feature detectors, local receptive fields of feature detectors etc. were inspired by the structure of the vision system of mammalian brains Lecun et al. (1998). There are also many differences between CNNs and these vision systems. A major difference is the preponderance of top-down or feedback connections in the biological vision Herzog et al. (2020b). It is believed that these feedback pathways are useful for many functionalities in the biological vision and that feedback is useful for stable visual perception Gilbert & Li (2013); Gilbert & Sigman (2007).

Motivated by this, many researchers have explored models of CNNs with feedback for many different applications. (See, e.g., Sam & Babu (2018); Yan et al. (2019); Cao et al. (2019). We briefly review these in the next subsection). By feedback in CNNs, we mean a mechanism whereby the output at higher layers of the network is used to generate some signals that would modify the output of the lower layers of the network in a dynamic or iterative fashion.

In this paper we present a novel mechanism for incorporating feedback in CNNs that we term as Image Specific Feature Selection (ISFS) which leverages feedback to dynamically select the most relevant filters for a given image. What we propose is a generic mechanism for incorporating feedback in any CNN used for a classification task.

In our model, the outputs of the last convolutional layer and the final (softmax) layer of the CNN are supplied to a separate feedback generator network which generates the feedback signals. The feedback signals would be binary vectors that are gated on to the outputs of the filters at different convolutional layers. This will result in the enabling of only a subset of the filters at different (convolutional) layers in the CNN. Since

the feedback signals are generated in response to each input image, this selection of filters would be image-specific. Since filters in a CNN represent feature detectors, we call this as image specific feature selection (ISFS). We want to point out that this is distinctly different from the general feature selection mechanisms, where we choose (or learn) a subset of relevant features for the classification problem as a whole. The idea of ISFS is that for each image, the features appropriate for that particular image should be used. All the feature detectors in the network may be relevant for the classification problem as a whole. But we have a feedback mechanism that would, for each given input image, results in a specific subset of features to be used for that image.

Considering a CNN used for object recognition, we can intuitively view this process as follows. Given the input image, the CNN would process it (in the usual feed-forward manner) to arrive at a tentative internal representation and a tentative decision on its category. Now using these current outputs of the network, feedback signals are generated that would disable some of the filters from being used at each layer of the CNN and the input image is processed again through this CNN. This is like bringing in the global context of this image in arriving at the final decision on this image.

We provide simulation results with multiple CNN architectures and multiple data sets to show that this feedback mechanism is interesting and is useful. We show that, in all cases, the feedback improves classification accuracy. We also show that feedback improves the performance of the network in terms of confidence in the classification as measured by Expected Confidence Error and Reliability diagrams Guo et al. (2017).Further, we provide some empirical evidence which suggests that our feedback mechanism results in sharper, and more semantically focused saliency maps.

Our empirical results show that the network indeed learns to make image-specific feature selection. The final learnt network uses only a subset of filters which is, by and large, distinct for each image. While more than 30% of the total filters (at the top convolutional layer) are discarded for any specific image, almost every filter is used for some image or the other.

At an anecdotal level, the image specific feature selection is intuitively satisfying. For example, consider the four images of the same class in each of the figures 1 and 2. In each figure, the two subsets of filters selected by the network for the first two images differ much more than those selected for the next two images. Visually, we can see that the first two images certainly have a higher level of dissimilarity compared to the next two images (even though all four belong to the same class).

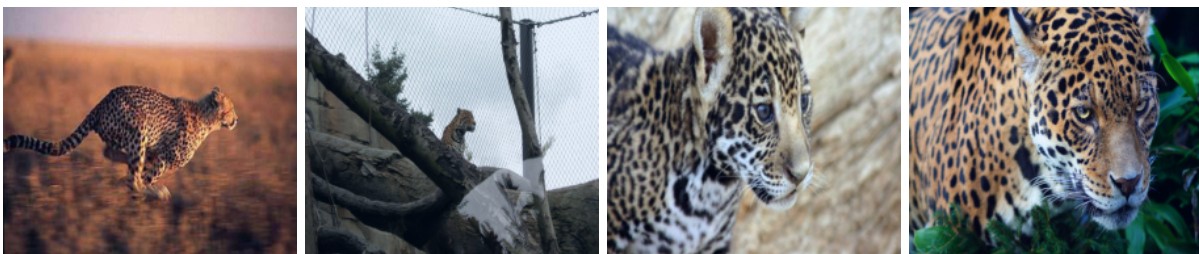

Figure 1: $1^{st}$ and $2^{nd}$ Image: Cheetah class images with binary feedback vectors at top convolution layer differing by 79 bits; $3^{rd}$ and $4^{th}$ Image: Images of Cheetah class whose binary feedback vectors at top convolution layer are identical. (Images from Imagenet-10).

There is another interesting aspect of this feedback structure. In a normal CNN classifier, the main inference or output available from the network for any given image is the outputs of the final softmax layer which are estimates of $P(c \mid X)$, where c is a class label and X is the input image. In addition, one can consider the output of the final convolutional layer (which can be viewed as the learnt internal representation of the image) as another output from the CNN. With our feedback mechanism, the binary feedback vector computed by the network for a given input image is an additional inference or output from the network. We can consider this as a low dimensional approximation to the computed internal representation (output of the final convolutional layer). The binary feedback vector implicitly contains some important global information present in the image. We explore this aspect in an interesting scenario. We give the trained network a

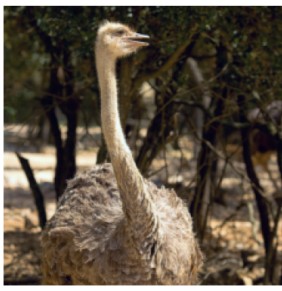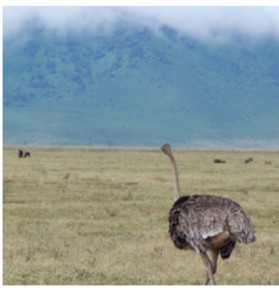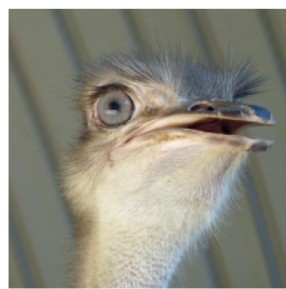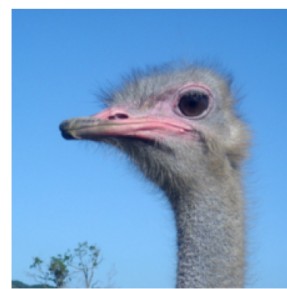

Figure 2: $1^{st}$ and $2^{nd}$ Image: Ostrich class images with binary feedback vectors at top convolutional layer differing by 82 bits; $3^{rd}$ and $4^{th}$ Image: Images of Ostrich class whose binary feedback vectors at top convolutional layer differing in 4 bits. (Images from Imagenet)

mix of normal as well as adversarially perturbed images and compute the binary feedback vectors. We now represent each image by its binary feedback vector (of dimension 256) and train an SVM classifier to distinguish between normal and adversarially perturbed images. This SVM achieves good accuracy even though each image is represented by only a 256-bit vector. We compare this classifier with an MLP classifier whose input is the output of the final convolutional layer in the CNN. This achieves a similar (though slightly less) accuracy. Thus the binary feedback vectors provide a good low-dimensional approximation to the learnt internal representation of images. We also show that the learnt SVM classifier can be used to get a CNN classifier with a reject option to be able to reject adversarially perturbed images with a high rate of success.

The main contributions of the paper can be summarized as follows. We propose a novel feedback mechanism in CNNs that we call Image-Specific Feature Selection. The method we propose is very generic and such a feedback can be added onto any CNN used as a classifier. The proposed feedback mechanism (through learning of the feedback generator network) results the system selecting a subset of relevant features for each image. We show that the feedback improves the performance of the network both in terms of accuracy and confidence in classification. We also show that using the feedback we get saliency maps that are more focussed on relevant parts of the object. We show that the feature selection resulting from the feedback is indeed image-specific and is interesting. We also explore the utility of the feedback signals in enabling the network to effectively reject input images that contain adversarial noise.

The rest of the paper is organized as follows. Section 1.1 discusses related work. Section 2 presents our proposed feedback mechanism. Section 3 discusses our empirical results, and concluding remarks are provided in Section 4.

## 1.1 Related Work

Feedback in CNNs has been explored for many different applications such as

- Object Classification : Nayebi et al. (2022); Herzog et al. (2020a); Huang et al. (2020); Kreiman & Serre (2020); Yan et al. (2019); Li et al. (2018); Nayebi et al. (2018); Nguyen et al. (2018); Wang et al. (2018); Zamir et al. (2017); Cao et al. (2015); Stollenga et al. (2014)

- Hyper-spectral Image Classification : Yu et al. (2021)

- Scene Parsing: Jin et al. (2017); Pinheiro & Collobert (2014); Liang et al. (2015)

- Segmentation : Tomar et al. (2022); Cao et al. (2019); Li et al. (2016)

- Pose Estimation : Belagiannis & Zisserman (2017); Carreira et al. (2016); Oberweger et al. (2015)

- Crowd Counting Sam & Babu (2018)

- Super Resolution Image Generation: Deng et al. (2021); Lee et al. (2020); Li et al. (2019)

- Saliency Detection : Ding et al. (2022)

- Object Localization : Kreiman & Serre (2020); Cao et al. (2019; 2015)

- Adversarial Improvement : Alamia et al. (2023); Huang et al. (2020); Yan et al. (2019)

- Text Prediction : Chung et al. (2015) etc.

In Li et al. (2018) and Stollenga et al. (2014), feedback generated using information from the top layers of the CNN, is used in an attention-like mechanism to improve performance in object recognition tasks. Similarly, Ding et al. (2022) presents an approach to improve saliency detection by selectively emphasizing regions that are most likely to attract human gaze. In Cao et al. (2015) and Cao et al. (2019), it is demonstrated that feedback can be used to selectively enhance certain regions in the lower layers of the network, which is useful for better object localization. In general, feedback in CNNs is seen to be effective in learning better representations across various vision tasks Zamir et al. (2017); Stollenga et al. (2014); Yan et al. (2019). In Nayebi et al. (2018), it is demonstrated that using local recurrent connections, a convolutional network with fewer layers can achieve recognition accuracies on the ImageNet dataset that are comparable to those of deep Residual Networks. Furthermore, Kreiman & Serre (2020) and Yan et al. (2019) highlight that modulating the output of adjacent layers through feedback can significantly improve the representation of lower layers.

Mostly, the different works cited above use different architectures and learning algorithms to implement feedback. In most cases, the proposed architecture for the CNN with feedback is rather application-specific. In contrast, the mechanism we propose here is quite generic. It is like an add-on module that can be used with any CNN. Also, the specific mechanism, namely image specific feature selection, that we propose here is novel and has not been explored in the literature.

In Huang et al. (2020) a general feedback based CNN is proposed where the the final representation in the forward CNN is used to recreate the input image through a generative model and the reconstructed image is once again passed through the forward CNN for a consistency check. While this is a general CNN with feedback, its feedback mechanism is limited to only reconstruction of input image (relying on a pixel-level reconstruction error for learning the generative model).

Another approach to feedback mechanisms in neural networks is the concept of predictive coding, as explored in Han et al. (2018). Predictive coding involves local recurrence within a visual area, rather than between adjacent visual areas. In Alamia et al. (2023), CNNs with predictive coding dynamics through feedback connections are shown to be effective for image classification, especially with noisy data.

## 2 CNN with Feedback for Image Specific Feature Selection

Consider a normal (feedforward) CNN with $L$ convolutional layers. We denote the output of layer $l$ by $y^l$. Then $y^l = (y_1^l, y_2^l, \ldots, y_{n_l}^l)$, where $n_l$ is the number of channels or filters at layer $l$ and each $y_c^l$ is an image of appropriate dimension. These outputs are calculated as

$$y^{l+1} = f_{l+1}(W^{l+1} \circledast y^l), \ l = 0, \cdots, L-1 \tag{1}$$

where $W^{l+1}$ is the weight tensor and $f_{l+1}$ is the activation function (e.g., ReLU) at this layer, and $\circledast$ symbolically represents the usual convolution operation done in CNNs to obtain output of any layer. We take $y^0$ to be the input image. In the CNN used for classification, these convolutional layers are followed by some fully connected layers with the final output layer being the softmax layer whose outputs are the estimated probabilities of different classes.

When we incorporate feedback, the outputs of the top layers of the CNN are used, through another network that we call *feedback generating* network (FBG), to generate feedback signals that modify the outputs of the lower layers. Since this process is iterative, let us denote by $y^l(k)$ and $m^l(k)$ the output and the feedback signal respectively at layer $l$ at $k^{th}$ iteration. Note that the input to the FBG that generates $m^l(k)$ would be $y^{l'}(k-1)$ for some values of $l' > l$. We will describe the network architecture and the learning algorithm in the next subsection.

Now we can write the equations for the output of layer $l + 1$ at $k^{th}$ iteration in the CNN with feedback as

$$y^{l+1}(k) = f_{l+1}(W^{l+1} \circledast (y^l(k) \odot m^l(k)))$$ (2)

where $\odot$ symbolically denotes the operation by which the feedback signal modifies the output of the layer. (We take $y^0(k)$ to be the input image for all $k$).

For us, the feedback signals would be binary and they would be gated onto the filters at each layer. Thus $m^l = (m_1^l, \cdots, m_{n_l}^l)$ is a binary vector of dimension $n_l$ which is the number of filters at layer $l$. Now we define the operation of the feedback by

$$y^l(k) \odot m^l(k) = (y_1^l(k) \otimes m_1^l(k), \cdots, y_{n_l}^l(k) \otimes m_{n_l}^l(k))$$ (3)

where $y_c^l(k) \otimes m_c^l(k)$ denotes the operation of multiplying each elements of the array $y_c^l(k)$ by the number $m_c^l(k)$. Note that $m_c^l(k) \in \{0, 1\}$. For a specific $c$, if $m_c^l = 0$, then that particular filter will not contribute any input to the next layer. Thus, only those filters whose corresponding $m_c^l$ are 1 are used at this layer. This is how the feedback results in image-specific selection of filters or features at each layer.

We take $m^l(1)$ to be a vector of all $1's$ so that $y^l(1) \odot m^l(1) = y^l(1)$ which will make the first iteration same as the normal pass through the (feed-forward) CNN. After calculating $y^l(1)$ for all layers, these are used to generate $m^l(2)$ for different $l$ (using the FBGs) which are then used in the above equations to get $y^l(2)$ for all layers and so on.

During Training, we need to decide on the number of times we iterate through eq.(2) for deciding on the final output of the network which would be used for computing the loss and updating the weights. The minimum number of iterations is 2 because, otherwise feedback has no effect. In all the experiments reported here, during training we used only two iterations. At inference time also we take the output of the second iteration as the final output of the network.

The overall system, thus, consists of a forward network which is a CNN and another network to generate the feedback which is the FBG. We discuss the architecture of the system in the next subsection.

**Remark 1**: This structure of a CNN with feedback is quite distinct from that of a general recurrent neural network (RNN). In an RNN, each layer is a dynamical system. In contrast, a CNN with feedback is essentially the coupling of two separate feed-forward networks, whereby after the initial pass through the forward network, signals travel backwards through another network so as to enable the system to iteratively refine its representations.

## 2.1 Architecture of the CNN with feedback

Our forward network will be a CNN with some $L$ convolutional layers which will be followed by some fully connected layers and a final softmax output layer whose outputs are the (estimated) probabilities of different classes.

For the CNN we choose standard off-the-shelf architectures. Our feedback mechanism is very generic and it can be incorporated into any CNN. Hence, for our experiments we choose standard off-the-shelf architectures for the CNN. We present results with ResNet-18, ResNet-34 He et al. (2016) and VGGNet Simonyan & Zisserman (2015). In the ResNets and the VGGNet, the convolutional layers are organized into groups. For the purpose of feedback we treat the group as a single entity and thus group $l$ would have a feedback vector, $m^l$. From now on, we refer to each group as a layer.

For each layer (group), $l$, we need to generate the feedback vector $m^l$. Thus, the number of FBGs corresponds to the number of layers receiving feedback.

The next question is what should be the structure of FBG network. The network structures employed for generating feedback signals are pretty diverse in the current literature on CNNs with feedback. Some authors used a second CNN Sam & Babu (2018), and some others use CNNs where the filters have fractional stride Huang et al. (2020); Yan et al. (2019). In contrast, some use simple feedforward MLPs Li et al. (2018). Some feedback-generating networks are quite non-standard Cao et al. (2019).

To keep the FBGs simple and generic, we choose a one hidden layer feedforward network (an MLP) for all the FBGs. The input to the FBGs is taken from the final convolutional layer and the softmax layer of the CNN. The output of the final convolutional layer is taken through a global average pooling (GAP) module to form one part of the input to the FBG while the other part is the output of the softmax layer. Thus the dimension of the input layer in the FBG is $n_L + M$ where $n_L$ is the number of filters in the final convolutional layer and $M$ is the number of classes. All FBGs share the weight matrix from input to the hidden layer. All hidden layer nodes in the FBG use ReLU activation. The output of FBG generating $m^l$ has dimension $n_l$. So, each FBG needs to have its own output layer with appropriate dimension. All nodes in the output layer of FBG use sigmoid activation function. We use a binarizer that thresholds these outputs at 0.5 to generate the final binary feedback vector.

Consider the system with the forward CNN being ResNet-18. In Resnet-18, the convolutional layers are grouped into five groups. Hence for us $L = 5$ here. In our system we supply feedback to three convolutional layers, namely groups 2, 3, and 4. These have, respectively, 64, 128, and 256 filters. We will have three FBGs each of which is a one-hidden-layer MLP. All of them share the input to hidden layer weights.

As mentioned earlier, we also experimented with configurations where the forward CNN is ResNet-34 or VGGNet.

ResNet-34 is also partitioned into five groups, so $L = 5$. As in the earlier ResNet-18 configuration, we inject feedback into three convolutional groups, namely, groups 2, 3, and 4, whose output stages contain 64, 128, and 256 filters, respectively. The architecture of the feedback generator (FBG) is kept identical to that used in the ResNet-18 setting.

VGG19 is also divided into five convolutional blocks ($L = 5$), with feedback injected before Blocks 2 to 5, which have 64, 128, 256, and 512 channels respectively. The feedback generator mirrors the ResNet-18/34 design.

Figure 3a illustrates the complete architecture when using ResNet-18 as the forward CNN, and Figure 3b shows the analogous setup with VGG-19. In each diagram, the shared part of the feedback generator is labeled `FB-base`, while the output layers which are different for different FBGs are labeled `FB-1`, `FB-2`, and `FB-3` (with an additional `FB-4` in the VGG-19 variant). Precise dimensions for the feedback vectors $m^l$ and the hidden layer sizes of these FBGs are provided in Section 3.

## 2.2 Learning Algorithm

The operation of this CNN with feedback is as follows. We present the input image at the input of the CNN. In the first iteration we take all feedback signals to be 1. Thus the first iteration would be the normal computation through a CNN using all the filters. Then we supply the output of last convolutional layer and the final softmax layer of the CNN as input to all the FBGs, which give us the binary feedback vectors $m^l(2)$ for iteration 2. Using these we now recompute the output of CNN using eq.(2). (As per this computation, in the second iteration, in each layer $l$, the output of filter $c$ is passed onto the next layer only when $m_c^l(2) = 1$, for $c = 1, \cdots, n_l$). Thus we get two outputs at the final softmax layer of the CNN. Both these would be vectors of (predicted) probabilities of different classes for the input image. Call the two vectors $q_1, q_2$. Here, $q_1$ would be for the case when we use all the filters in the CNN while $q_2$ is in the case where we use only those filters selected by the feedback signals. At test time, we use $q_2$ as the final output from the CNN. For training the CNN and the FBGs we follow the procedure described below.

For each input training image we compute the outputs $q_1$ and $q_2$. Suppose the class label of the training image is $\tilde{y}$ which we take to be a one-hot vector. Then we first learn the CNN weights using $\text{CCE}(q_1, \tilde{y})$ as the loss function where the general CCE loss function is defined below

$$\text{CCE}(t, s) = -\sum_{i=1}^{M} t_i \log(s_i) \tag{4}$$

Note that during this learning of CNN, we are not using the FBGs.

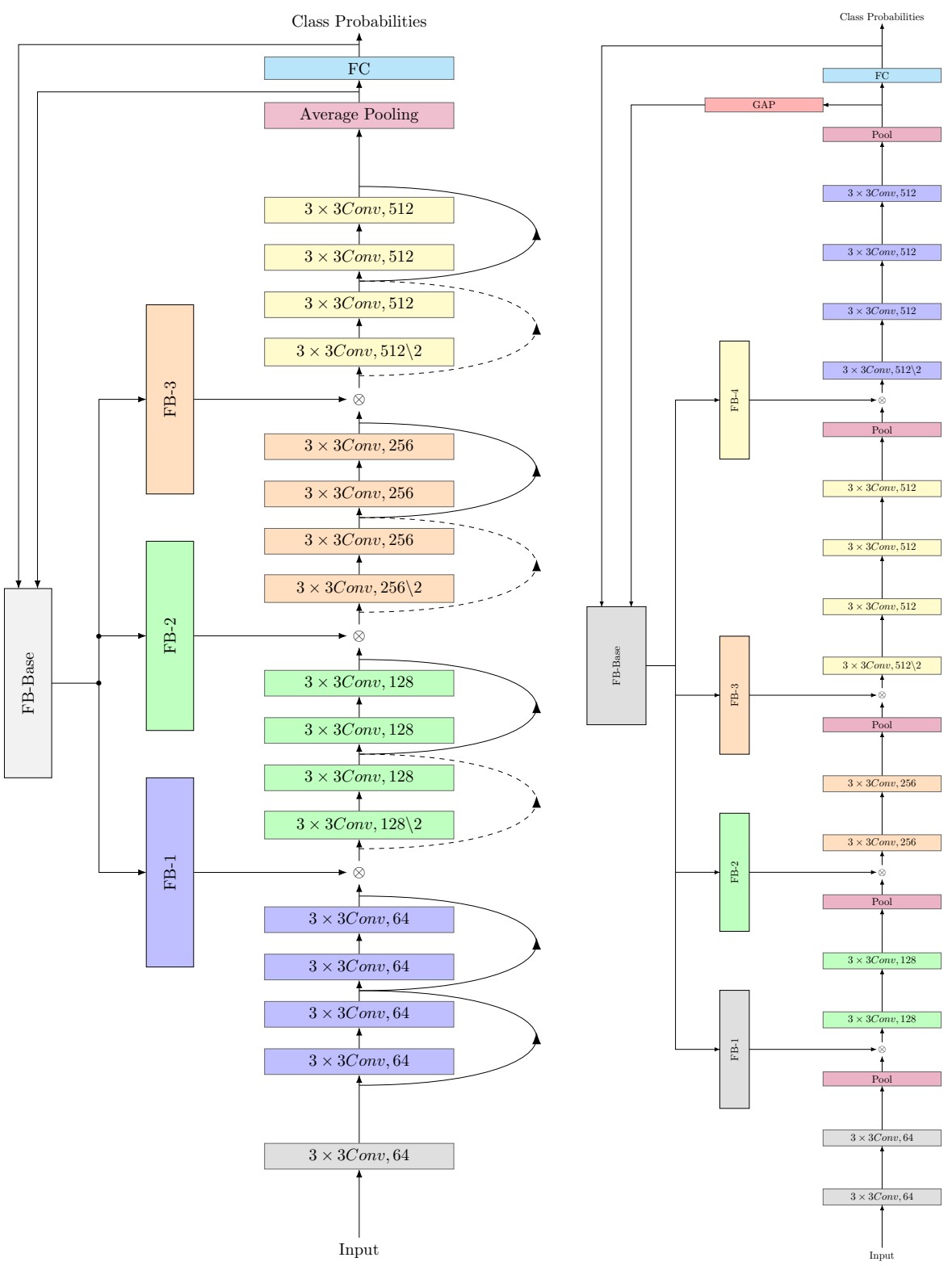

(a) The architecture of the ResNet-18 with feedback    (b) The architecture of the VGG-19 with feedback

Figure 3: Architectures of our feedback-augmented CNNs: (a) ResNet-18 variant and (b) VGG-19 variant, each injecting learned binary gating vectors at multiple blocks.

After 30 epochs of learning the CNN, we freeze the weights in the CNN and then learn the weights in the FBGs using $CCE(q_2, \tilde{y})$ as the loss function. In the FBGs we have a binarizer and we need to backpropagate the error through it for learning the weights. For the purposes of backpropagation we treat the binarizer as an identity map.

There are two reasons for this learning protocol. The feedback here is intended to select a subset of filters that are relevant for a given image. The FBG network is to be trained so that it learns to select appropriate filters. For such learning it would be good if the filters in the CNN are stabilized. That is why we first train the CNN separately and then learn the weights in the FBG network. A second reason for this protocol is that we want to think of ISFS as a general pupose mechanism that can be used with any CNN. That is why we take a CNN that is trained separately and then incorporate the feedback and train only the weights in the FBGs. As a matter of fact, in our experiments involving the 1000-class Imagenet data, we download a publicly available CNN for this data set and then train only the FBGs. We also note here that, during training of the FBG, the number of weights to be learnt is only about 1% of the number of weights in the Resnet. Thus adding our feedback mechanism to a CNN has negligible overheads in terms of the additional weights that are added.

## 3 Empirical Results

In this section, we present the experimental results obtained with our feedback mechanism of image-specific feature selection. We show that the feedback improves the performance of the network in terms of both classification accuracy, and Expected Confidence Error Guo et al. (2017). We also show that it can result in better Saliency maps which computes a pixel-wise importance. We demonstrate that the binary feedback vectors generated are indeed image-specific and that they are a good low-dimensional approximation to the internal representation of the image.[1]

### 3.1 Experimental Setup

**Forward Network:** As mentioned earlier, we used three different widely used CNNs: ResNet-18 and ResNet-34 He et al. (2016), and VGG-19 Simonyan & Zisserman (2015). ResNet-18 comprises 17 convolutional layers organized into five residual blocks (with a single convolution layer in Block 1), while ResNet-34 extends this design to 33 layers across the same block structure. VGG-19 consists of 19 weight layers arranged into five convolutional blocks. In both ResNet variants, we apply our feedback mechanism at the outputs of Blocks 2, 3, and 4, which have 64, 128, and 256 feature channels, respectively. In VGG-19 we inject feedback at the outputs of Blocks 1 to 4, corresponding to 64, 128, 256, and 512 channels.

**Feedback Generator:** As mentioned earlier, each of our FBGs is an MLP with one hidden layer. The dimension of the input layer is $512 + k$(where 'k' is the number of classes) and the the output of the final convolutional layer (or block) in all our forward network architectures would have dimension 512 after global average pooling. The hidden layer has 128 nodes and the input to hidden layer weights are shared by all FBGs. (This part is shown as FB-base in Figure 3). The number of nodes in the output layer of an FBG is the same as the number of filters in the layer to which this feedback signal is sent. Figure 3 illustrates the overall architecture when we use ResNet-18 and VGG-19 as forward network. The total number of weights in all the FBGs put together is about 1% of the weights of the forward network. Thus introduction of feedback does not add any significant computational complexity.

**Data sets:** We evaluate our method on both small and large-scale classification benchmarks. First, we construct five disjoint 10-class subsets from ImageNet – denoted ImageNet-10(D1) through ImageNet-10(D5) – each containing 1,300 training and 100 test images per class. Second, we test on several balanced and imbalanced datasets with much larger numbers of categories: Caltech-101 (101 classes, 31 to 800 images per class), Caltech-256 (256 classes, 80 to 800 images per class), Flowers-102 (102 classes, 40 to 250 images per class), Tiny ImageNet (200 classes, 500 training images per class), and the full ImageNet-1K (1,000 classes). (We get similar results on the simple data sets such as MNIST and Fashion-MNIST; but these are not shown in the results presented here.).

---

[1]Codes will be made available post-acceptance

**Training Procedure:** We described the learning method in Section 2.2. We first train the CNN alone for 30 epochs. After that we freeze the weights in the CNN. Subsequently, we train only the feedback generating networks for 70 epochs.

We compare the performance of the CNN with and without feedback. For incorporating feedback, we first train the forward CNN alone for 30 epochs. Then we freeze the CNN weights and train only the FBGs.[2]. For obtaining the performance of the CNN without feedback, we train the CNN for about 50 to 100 epochs. We train the CNN till the training loss stabilizes. We do not provide comparisons with any other model of CNN with feedback. Almost all the reported architectures of CNN with feedback are rather application specific. Our mechanism for feedback is fairly generic. It can be added onto any pretrained CNN. Further, the feedback mechanism we proposed here is novel. The simulations are intended to demonstrate the performance enhancements this novel feedback can bring about.

## 3.2 Prediction Accuracy of the CNN with feedback

The classification accuracy of the trained network, with and without feedback, is shown in Table 1. The results shown in the table are averages (with standard deviation) over five repetitions.[3] As can be seen from the tables, across all the different data sets and for all the three CNN architectures, Feedback improves the accuracy and the increase is significant given the standard deviation.

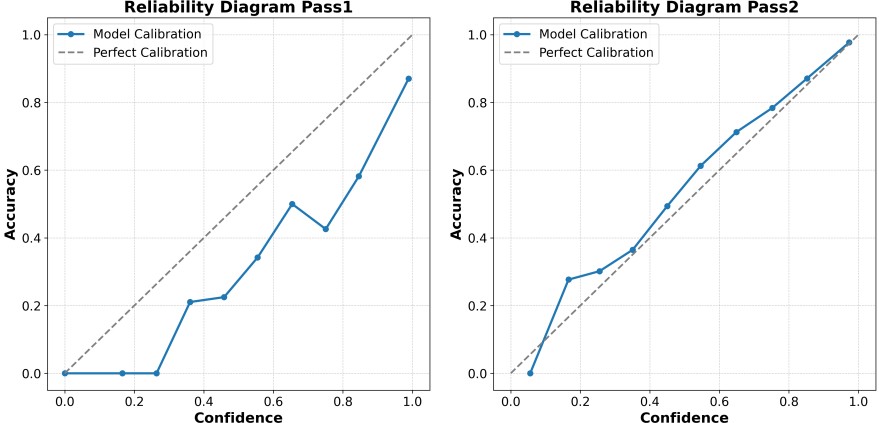

Figure 4: Reliability Diagrams for Flowers 102 using ResNet-18: without feedback (left panel) and with Feedback (right panel)

## 3.3 Confidence Error and Reliability Diagrams

Apart from percentage accuracy, another way to measure the performance of a classifier is through measures characterizing the confidence. Reliability diagrams and confidence errors are tools for evaluating probabilistic classification modelsGuo et al. (2017). The confidence error quantifies the misalignment between a model's predicted confidence and the actual accuracy.

*Expected Confidence Error* (ECE) evaluates how well a probabilistic classifier's predicted confidence aligns with true accuracy Guo et al. (2017). For a sample $i$, let $p_i = \max_k \hat{p}_i^k$, where $\hat{p}_i^k$ is the predicted probability for class $k$. Predictions are grouped into bins $B_m$, $m = 1, \cdots, M$, such that sample $i$ is put in $B_m$ if $p_i \in (\frac{m-1}{M}, \frac{m}{M}]$. For each bin $B_m$, accuracy and average confidence are calculated as:

$$\text{acc}(B_m) = \frac{1}{|B_m|} \sum_{i \in B_m} \mathbb{1}_{\hat{y}_i = y_i}, \quad \text{conf}(B_m) = \frac{1}{|B_m|} \sum_{i \in B_m} p_i$$

---

[2]Only in case of Imagenet-1000 data set, we use a publicly available CNN model and we do not train the forward CNN

[3]In the tables the standard deviation is not shown if it is less than 0.001.

Table 1: Predictive accuracy (**mean ± std**) across datasets without and with Feedback

| Model | Dataset | Without Feedback | With Feedback |
|---|---|---|---|
| **ResNet-18** | Imagenet-10 (D1) | 0.92 ± 0.005 | 0.94 ± 0.003 |
| | Imagenet-10 (D2) | 0.94 ± 0.006 | 0.96 ± 0.004 |
| | Imagenet-10 (D3) | 0.9 ± 0.008 | 0.92 ± 0.004 |
| | Imagenet-10 (D4) | 0.88 ± 0.008 | 0.91 ± 0.005 |
| | Imagenet-10 (D5) | 0.95 ± 0.004 | 0.97 ± 0.003 |
| | Caltech-101 | 0.883 ± 0.012 | 0.902 ± 0.008 |
| | Caltech-256 | 0.678 ± 0.006 | 0.689 ± 0.007 |
| | Flowers-102 | 0.943 ± 0.021 | 0.958 ± 0.03 |
| | Tiny ImageNet | 0.71 ± 0.009 | 0.725 ± 0.005 |
| | ImageNet 1K | 0.673 | 0.681 |
| **ResNet-34** | Imagenet-10 (D1) | 0.93 ± 0.005 | 0.943 ± 0.008 |
| | Imagenet-10 (D2) | 0.941 ± 0.006 | 0.965 ± 0.009 |
| | Imagenet-10 (D3) | 0.912 ± 0.007 | 0.928 ± 0.003 |
| | Imagenet-10 (D4) | 0.895 ± 0.005 | 0.918 ± 0.006 |
| | Imagenet-10 (D5) | 0.954 ± 0.004 | 0.968 ± 0.008 |
| | Caltech-101 | 0.901 ± 0.008 | 0.916 ± 0.005 |
| | Caltech-256 | 0.695 ± 0.003 | 0.718 ± 0.007 |
| | Flowers-102 | 0.954 ± 0.007 | 0.97 ± 0.006 |
| | Tiny ImageNet | 0.728 ± 0.005 | 0.732 ± 0.006 |
| | ImageNet 1K | 0.725 | 0.731 |
| **VGG-19** | Imagenet-10 (D1) | 0.89 ± 0.011 | 0.925 ± 0.001 |
| | Imagenet-10 (D2) | 0.918 ± 0.017 | 0.938 ± 0.005 |
| | Imagenet-10 (D3) | 0.91 ± 0.009 | 0.927 ± 0.002 |
| | Imagenet-10 (D4) | 0.869 ± 0.008 | 0.888 ± 0.005 |
| | Imagenet-10 (D5) | 0.926 ± 0.011 | 0.939 ± 0.012 |
| | Caltech-101 | 0.848 ± 0.005 | 0.859 ± 0.002 |
| | Caltech-256 | 0.63 ± 0.008 | 0.642 ± 0.003 |
| | Flowers-102 | 0.883 ± 0.005 | 0.905 ± 0.006 |
| | Tiny ImageNet | 0.675 ± 0.005 | 0.677 ± 0.002 |
| | ImageNet 1K | 0.69 | 0.701 |

where $\hat{y}_i$ and $y_i$ are the predicted and true labels, respectively. The ECE is:

$$\text{ECE} = \sum_{m=1}^{M} \frac{|B_m|}{n} \left| \text{acc}(B_m) - \text{conf}(B_m) \right|,$$

where $n$ is the total number of samples. Lower ECE implies better calibration. Ideally we want $\text{acc}(B_m) \approx \text{conf}(B_m)$ for all $m$.

*Reliability diagram* visually represents this relationship by plotting $\text{acc}(B_m)$ (y-axis) against $\text{conf}(B_m)$ (x-axis) for each bin. Perfect calibration corresponds to the diagonal line $\text{acc}(B_m) = \text{conf}(B_m)$. Deviations from

Table 2: Expected Confidence Error (ECE) with and without Feedback across datasets and architectures

| Model | Dataset | Without Feedback | With Feedback |
|---|---|---|---|
| **ResNet-18** | ImageNet-10 (D1) | $0.021 \pm 0.003$ | $0.003 \pm 0.0008$ |
| | ImageNet-10 (D2) | $0.014 \pm 0.004$ | $0.002 \pm 0.0007$ |
| | ImageNet-10 (D3) | $0.031 \pm 0.005$ | $0.003 \pm 0.0008$ |
| | ImageNet-10 (D4) | $0.028 \pm 0.007$ | $0.005 \pm 0.0011$ |
| | ImageNet-10 (D5) | $0.018 \pm 0.003$ | $0.004 \pm 0.0019$ |
| | Caltech-101 | $0.053 \pm 0.018$ | $0.005 \pm 0.005$ |
| | Caltech-256 | $0.125 \pm 0.026$ | $0.022 \pm 0.002$ |
| | Flowers-102 | $0.054 \pm 0.006$ | $0.005 \pm 0.002$ |
| | Tiny ImageNet | $0.121 \pm 0.016$ | $0.07 \pm 0.002$ |
| | ImageNet 1K | $0.0377$ | $0.0214$ |
| **ResNet-34** | ImageNet-10 (D1) | $0.025 \pm 0.011$ | $0.007 \pm 0.002$ |
| | ImageNet-10 (D2) | $0.023 \pm 0.005$ | $0.006 \pm 0.004$ |
| | ImageNet-10 (D3) | $0.027 \pm 0.005$ | $0.005 \pm 0.0006$ |
| | ImageNet-10 (D4) | $0.032 \pm 0.005$ | $0.006 \pm 0.0005$ |
| | ImageNet-10 (D5) | $0.016 \pm 0.004$ | $0.005 \pm 0.0002$ |
| | Caltech-101 | $0.065 \pm 0.002$ | $0.003 \pm 0.002$ |
| | Caltech-256 | $0.045 \pm 0.005$ | $0.005 \pm 0.008$ |
| | Flowers-102 | $0.033 \pm 0.015$ | $0.006 \pm 0.005$ |
| | Tiny ImageNet | $0.172 \pm 0.012$ | $0.092 \pm 0.004$ |
| | ImageNet 1K | $0.0438$ | $0.0328$ |
| **VGG-19** | ImageNet-10 (D1) | $0.039 \pm 0.005$ | $0.016 \pm 0.008$ |
| | ImageNet-10 (D2) | $0.058 \pm 0.012$ | $0.033 \pm 0.007$ |
| | ImageNet-10 (D3) | $0.040 \pm 0.005$ | $0.010 \pm 0.004$ |
| | ImageNet-10 (D4) | $0.050 \pm 0.017$ | $0.016 \pm 0.009$ |
| | ImageNet-10 (D5) | $0.026 \pm 0.008$ | $0.017 \pm 0.009$ |
| | Caltech-101 | $0.073 \pm 0.006$ | $0.004 \pm 0.002$ |
| | Caltech-256 | $0.056 \pm 0.004$ | $0.005 \pm 0.001$ |
| | Flowers-102 | $0.062 \pm 0.003$ | $0.012 \pm 0.010$ |
| | Tiny ImageNet | $0.235 \pm 0.013$ | $0.150 \pm 0.021$ |
| | ImageNet 1K | $0.124$ | $0.091$ |

the diagonal indicate overconfidence (accuracy < confidence) or underconfidence (accuracy > confidence). These metrics enable a rigorous assessment of model reliability.

We evaluated these metrics both before and after applying feedback. Table. 2 shows the ECE without and with feedback. As we can see there is a significant reduction in the ECE when we use feedback.

Figure 4 ,5 shows the reliability diagram for the data set Flowers-102 and Imagenet-10(D5) respectively, which illustrates that the model with feedback is better calibrated compared to the model without feedback. Similar results are observed for other data sets also. (More such images can be seen in supplementary material) These results demonstrate that our feedback mechanism improves the performance of the model in terms of confidence scores and reliability of the model also.

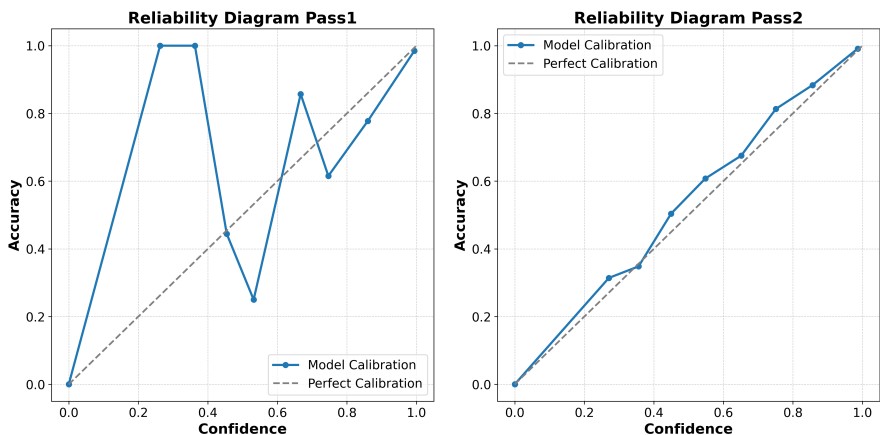

Figure 5: Reliability Diagrams for Imagenet-10(D5) with VGG-19: without feedback (left panel) and with Feedback (right panel)

Table 3: Number of distinct binary feedback vectors generated at different blocks for different classes of Imagenet-10(D1) with ResNet-18 as the Forward Network

| Block | C-0 | C-1 | C-2 | C-3 | C-4 | C-5 | C-6 | C-7 | C-8 | C-9 | Overall |
|-------|-----|-----|-----|-----|-----|-----|-----|-----|-----|-----|---------|
| Block-2 | 136 | 259 | 129 | 120 | 221 | 305 | 184 | 221 | 231 | 130 | 1368 |
| Block-3 | 949 | 1096 | 632 | 729 | 780 | 749 | 955 | 970 | 712 | 762 | 8285 |
| Block-4 | 1248 | 1281 | 1224 | 1206 | 1271 | 1236 | 1265 | 1175 | 1224 | 1222 | 12270 |

### 3.4 Feedback-guided Salience Analysis

Saliency analysis computes a pixel-wise importance map by taking the magnitude of the input gradient $|\nabla_{\mathbf{x}} s_c(\mathbf{x})|$(Here, $\mathbf{x}$ denotes the input image tensor, $s_c(\mathbf{x})$ the model's output score for class $c$, $\nabla_{\mathbf{x}} s_c(\mathbf{x})$ its gradient with respect to $\mathbf{x}$, and $|\cdot|$ the element-wise absolute value). This quantity helps in identifying regions of an image that most strongly influence the predicted class score. As a direct, post-hoc visualization of the network's "attention", such maps help verify that decisions rest on semantically meaningful features rather than spurious cues, and it can expose failure modes (for example, over-focus on background artifacts). Such an analysis is often useful in dataset curation and architecture refinement, and can help evaluate the model's trustworthiness.

We investigated the utility of feedback in terms of such a saliency analysis. For each image, we generate $|\nabla_{\mathbf{x}} s_c(\mathbf{x})|$ as a heat-map. We do this both with and without feedback. In the no-feedback case, the full gradient of the class score propagates back through every filter, to produce the heatmaps; in the case with-feedback, only those convolutional filters selected by the binary feedback vector are permitted to transmit the gradient while generating the heatmap. We present these results in Figure 6. In the figure we show some sample saliency maps of Imagenet-10 images using the Resnet-18 CNN with and without feedback. The results demonstrate that feedback helps concentrate saliency on the most informative pixels, aligning more closely with human intuition about key object parts and thus providing a basis for better interpretability. These results suggest that our image-specific selection of features through feedback seems to be helping the network choose more semantically relevant features for an image. These results are only indicative and they suggest that it is worthwhile to explore the utility of this feedback mechanism for generating better explanations. A full exploration of this issue is beyond the scope of this paper.

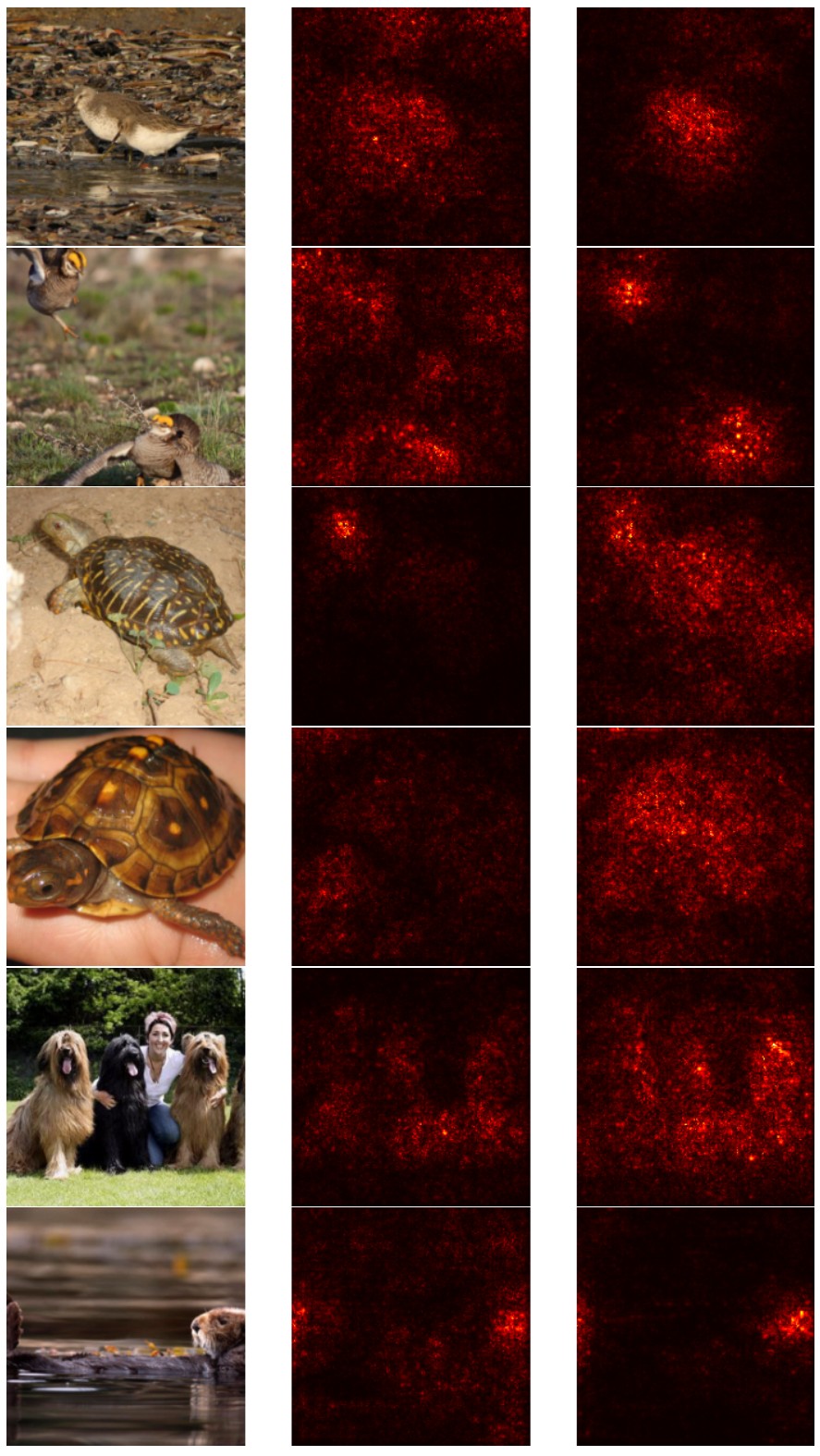

Figure 6: Saliency comparison on ImageNet-10 subsets using ResNet-18: each row presents (left to right) the original image, the saliency map without feedback, and the feedback-filtered saliency map. The maps obtained by using the feedback signals are noticeably sharper and more focused on the object's core regions.

Table 4: Number of distinct binary feedback vectors generated at different blocks for different classes of Imagenet-10(D4) with ResNet-18 as Forward Network

| Block | C-0 | C-1 | C-2 | C-3 | C-4 | C-5 | C-6 | C-7 | C-8 | C-9 | Overall |
|---|---|---|---|---|---|---|---|---|---|---|---|
| Block-2 | 219 | 89 | 294 | 221 | 126 | 105 | 210 | 399 | 177 | 243 | 1567 |
| Block-3 | 924 | 589 | 919 | 745 | 767 | 644 | 1043 | 631 | 798 | 839 | 7813 |
| Block-4 | 1268 | 1235 | 1257 | 1269 | 1261 | 1166 | 1297 | 1263 | 1201 | 1257 | 12236 |

Table 5: Variation in the number of filters used on ImageNet-10 (D2) with ResNet-18 as the forward network.

| | # Common Filters Used in Every Image | # Filters Used Across All Images | Avg. % of Filters Used per Image |
|---|---|---|---|
| Block–2 | 35 | 64 | 92 |
| Block–3 | 54 | 128 | 85 |
| Block–4 | 77 | 250 | 69 |

### 3.5 Analysis of the Binary Feedback Vectors

The feedback signal at each level is in the form of a binary vector that decides which all filters are used at that level for the current image. We analyze the image-specific nature of the feedback by calculating how many distinct binary vectors are generated at different blocks of the Forward network. In this and the next section we are providing these results where ResNet-18 is used as forward network. (Similar results are obtained for other networks and datasets, which are given in the supplementary material). In ResNet-18, feedback is delivered to blocks 2,3, & 4, and these blocks have 64, 128 and 256 filters respectively.

Tables 3 and 4 show the number of distinct binary feedback vectors at blocks 2, 3, and 4 (separately for each class as well as for all the classes combined) for the Imagenet-10(D1) and Imagenet-10(D4) data sets respectively. These results show that higher blocks have more distinct binary vectors. This is intuitively clear because lower convolutional layers would be learning basic feature detectors (which are needed for all images) while more complex feature detectors are learnt in deeper convolutional layers. As can be seen from the table, at block 4, more than 12,000 different binary vectors are generated in response to the 13,000 images. These results show that the feedback vectors are very much image-specific at block 4.

By considering all the distinct binary vectors generated at a level we can calculate the number of filters at that level used by every image and the number of filters used by at least one image (by bitwise AND and OR operations on the ensemble of the distinct binary vectors). These are shown for Imagenet-10(D2), in Table 5. The table also shows the average number of filters used per image. At block 4, on the average, less than 70% of the filters are used by any specific image. However, almost 250 (out of 256) filters are used for some image or the other.

Table 6: Confusion Matrix for Normal Vs Adversarial classification for Imagenet-10(D1) (and Imagenet-10(D3) in the brackets) with ResNet-18 as Forward Network

| | | Predicted Label | |
|---|---|---|---|
| | | Normal | Adversarial |
| Actual Label | Normal | 88.6% (87.83%) | 11.4% (12.17%) |
| | Adversarial | 9.7% (9.84%) | 90.3% (90.16%) |

We can get some intuitive idea of this image-specific feature selection by looking at pairs of images of the same class that result in similar or dissimilar feedback vectors. As mentioned in Sec. 1, such pairs of images are shown in figures 1 & 2. (More such images are given in the supplimentary material). In these figures the first two images in the row are dissimilar even though they belong to the same class. The system chooses fairly different subsets of filters for them. This is in contrast to the next two images in the row which are similar and the system chooses almost identical subset of filters for them. One can see that our intuitive idea of similarity of images is reflected by the similarity of feedback vectors.

We did an ablation experiment to see the utility of feedback in selecting a *relevant* subset of filters. For each image we generate the feedback vector and count the number of 1's in it. Then we generate a random binary vector with the same number of 1's and use it for feature selection. For the ImageNet-10 (D5) dataset, this approach causes the accuracy to drop from 0.966 to 0.71. This demonstrates that the selected features are indeed relevant for the image, as any random combination of the same number of features does not yield the same accuracy.

### 3.6 Classifying the image as Plain or Adversarial using their Binary Vectors

Since the feedback vectors are calculated using the internal representation of the image at the final convolutional layer, they contain some information about the image as a whole. We next explore whether the global context of image that is possibly captured by the binary feedback vectors is good enough for distinguishing between normal and adversarial images. Given the trained network we generate adversarial examples using both FGSM and PGD methods. We process the normal as well as adversarial images through the system and represent each image by the (256-bit) binary feedback vector it generates at block 4. An SVM classifier with a linear kernel is learnt to classify the binary vectors as normal or adversarial.

The learnt SVM classifiers deliver good performance in distinguishing between normal and adversarial images with accuracies on test set varying from 0.91 to 0.85 for the five Imagenet-10 datasets.

We also trained a single hidden layer MLP (with 100 hidden nodes), whose input is the output of the final convolutional layer (after passing through Global Average pooling layer), for distinguishing between plain and adversarially perturbed images. This classifier gives accuracy of 0.85 on the test set which is comparable to the accuracy achieved with the binary feedback vectors. Thus, we can say that the binary feedback vectors indeed capture enough information from the internal representation of the image.

To evaluate the effectiveness of this view of the binary feedback vectors, we use the learnt SVM classifier along with our trained CNN with feedback as follows. For each test image, we generate the binary feedback vector using the CNN with feedback and send it to the SVM to get a normal or adversarial classification. The Confusion Matrix for this classification is shown in Table 6 for the FGSM & PGD attacks for Imagenet-10(D1) and Imagenet-10(D3). As can be seen from the table, we are able to correctly identify adversarial images about 90% of the time. (However, for 11% of the time we wrongly flag a normal image as adversarial). Under this experimental design we obtain an accuracy of 0.9, precision of 0.82, recall of 0.89 and F1-score of 0.85. This shows that we can use the CNN with feedback along with the SVM, as a classifier with a reject option to be able to reject input images that may be adversarially perturbed. More importantly, this suggests that the binary feedback vectors indeed capture some useful global information about the image which may be exploited for different purposes.

## 4 Conclusion

In this paper, we presented a novel feedback mechanism for CNNs that we called image-specific feature selection. This is a generic mechanism that can be used with any CNN in image classification tasks. We can incorporate it into any pretrained CNN by training only the feedback generating networks. The FBGs are very simple MLPs and the additional computational complexity by adding the feedback is very small. (In our experiments, the weights in the FBGs are less than 1% of the weights in the CNN).

What we proposed is an interesting feedback mechanism that enables the network to learn to dynamically choose a subset of features tailored for each individual image. Using multiple data sets and different CNN

architectures, we demonstrated that the proposed feedback mechanism improves the performance in terms of both classification accuracy as well as confidence & reliability scores and saliency. Our experiments used data sets where number of classes varied over 10, 100, 200, and 1000. Some data sets also had severe class imbalance. But in all cases the accuracy and confidence scores showed significant improvement. Our empirical results also show that the binary feedback vectors are very much image specific. We also illustrated through an example that, among images of the same class, the similarity between the feedback vectors does reflect our intuitive notion of similarity between images. The binary feedback vectors generated for any given image through this mechanism contain some implicit global information about the image. We illustrated the utility of this by showing that this information can be used through a simple SVM classifier for distinguishing between normal and adversarial images with a good level of accuracy.

The feedback mechanism proposed here is very generic and can be used with any CNN. The feedback is helping the system to select features that may be particularly relevant for the given image. Our experiments clearly demonstrate the potential of this feedback mechanism More work is needed to explore this feedback mechanism further to fully realize the potential of CNNs with feedback. One issue with our feedback mechanism is that we use the feedback only for one iteration. One needs to explore stability of such feedback. While it may be difficult to get useful theoretical results on stability, it would be interesting to explore mechanisms for learning FBGs that promote stability. This is an important direction in which the proposed feedback idea can be explored further.

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
