# OpenReview forum: "Exploring a novel Feedback Mechanism for Convolutional Neural Networks"
_TMLR — Rejected by TMLR_

### Review · Reviewer_D8XH · 2025-08-22

**Summary Of Contributions:**

This paper adds a feature selection feedback mechanism to CNNs. It shows that this feedback increases accuracy, encourages more correlation between accuracy and confidence, and makes saliency maps look more reasonable.

**Additional Comments:**

The paper is clearly written but also very repetitive, which makes it tedious to read. Also there is a sentence in red that should be fixed.

**Audience:**

Yes

**Audience Explanation:**

While the feedback method itself is not terribly novel, I think the analysis in terms of confidence curves, looking at the sparsity of feature maps chosen, and impact on saliency maps is interesting.

**Claims And Evidence:**

Yes

**Claims Explanation:**

The claims are mainly about the impacts of the feedback and these impacts are demonstrated in the analyses. However, the saliency map analysis appears anecdotal (as the authors note).

The claim that this feedback method is novel, however, is not supported. As the authors cite, many previous works have added similar styles of feedback. Additional missing citations have as well:
[1] https://konklab.fas.harvard.edu/Papers/Konkle_2023_Neurips.pdf
[2] https://www.biorxiv.org/content/biorxiv/early/2022/03/08/2022.03.07.483196.full.pdf
The form of feedback here is therefore a variant on a theme, not a new method.

**Requested Changes:**

Regarding the novelty claim, the authors should downplay claims to novelty and instead specify exactly how their method fits into the existing set of feedback methods, and explain why they chose the details that they did and what the impacts were. For example, the authors use binary gating* rather than graded multiplicative scaling (as is commonly used). What are the motivations for and impact of this? Is this the main novel detail that they want to highlight (though this too has been done before, e.g. [3]) and compare to past models?

[3] https://openaccess.thecvf.com/content_CVPR_2019/papers/Chen_You_Look_Twice_GaterNet_for_Dynamic_Filter_Selection_in_CNNs_CVPR_2019_paper.pdf

*I believe this statement indicates that the model is not trained with binary gating variables, but rather uses the raw output from the sigmoid: "For the purposes of backpropagation we treat the binarizer as an identity map". But then at test-time these sigmoid outputs are binarized. What is the performance impact of this change at test time? This should be addressed in the paper.

The authors also claim that prior feedback methods are 'application-specific' unlike theirs. I do not know what this claim means or how it is supported. Almost all the feedback work is demonstrated on similar if not the same classification datasets used here and has the same level of generality. Other work also studies 'add-on' methods that can be added to any network (e.g. Predify, and ref [2] above).

I found it strange that the authors reported on the ability to predict adversarial images with the feedback method but did not report on if feedback helps with performance on adversarial images. It would be good to include those results as well.

This is not true and should be removed: "Predictive coding involves local recurrence within a visual area, rather than between
adjacent visual areas." Canonically, predictive coding requires top-down inputs as the predictions, and passes forward errors (Rao & Ballard)

---

### Review · Reviewer_Si6k · 2025-08-25

**Summary Of Contributions:**

This paper proposes a new technique for Convolutional Neural Networks (CNNs) called Image-Specific Feature Selection that can be added to any existing CNN used for classification. The ISFS introduces adaptable feedback loop for CNNs that makes them more accurate, transparent, and secure by forcing them to focus only on the most important features for each specific image.

**Audience:**

Yes

**Audience Explanation:**

People in the filed of Vision backbone design may be interst in this paper.

**Claims And Evidence:**

Yes

**Claims Explanation:**

The proposed ISFS show performance gain compared to baseline.

**Requested Changes:**

The paper, in its current form, has several major weaknesses that limit its impact.

1.  **Presentation Could Be Improved:** The writing and organization of this paper could be significantly improved.
    *   **Redundant Introduction:** The introduction for the proposed ISFS module is redundant, making it difficult to follow.
    *   **Poor Organization of Tables and Figures:** Many of the figures are poorly presented. For example, Fig. 1 and Fig. 2 could be merged as they serve the same purpose. Additionally, some figures occupy too much space and do not clearly highlight the proposed module. Fig. 3 takes up an entire page but could be made much more concise.
    *   **Inconsistent Symbols and Writing:** The notation is inconsistent. For instance, the related work section uses three different formatting styles: "Scene Parsing:", "Segmentation :", and "Crowd Counting". Furthermore, the symbol $k$ is used in both Sec. 2 and Sec. 3 to denote different meanings.

2.  **Limited Empirical Evaluation:** The experimental results provided are not comprehensive enough.
    *   **Marginal Gains:** The performance improvement over the baseline is minimal, especially on large-scale datasets like ImageNet.
    *   **Lack of Detailed Ablation:** Key design choices are not justified. For example, the choice of $K_{th}$ is not evaluated through an ablation study or otherwise explained.
    *   **Efficiency:** The impact of the proposed module on inference efficiency is not reported.
    *   **Generalization:** Although the authors claim the module can be integrated into any CNN classifier, results are only shown for ResNet and VGG. To substantiate this claim, experiments on more advanced models, such as ConvNeXt, would be beneficial.

---

### Review · Reviewer_yFkT · 2025-11-30

**Summary Of Contributions:**

The paper proposes a novel feedback mechanism termed "Image Specific Feature Selection (ISFS)" and corresponding "Feedback Generator (FBG)" network. The authors demonstrate that this method improves classification accuracy and calibration (ECE) on datasets.

**Additional Comments:**

Please see the part of requested changes.

**Audience:**

Yes

**Audience Explanation:**

The paper really design a feedback mechanism with the FFN, which is different from the current ML architecture, so it sounds like interesting and highly relative.

**Claims And Evidence:**

Yes

**Claims Explanation:**

The authors provide the empirical evidence about the method, and the claimed points I believe are not accessible in some scenarios. So I think in provided scenes, the conclusion is right.

**Requested Changes:**

I think there are still lots of things that need to be done by authors.
The main weakness:
1. The novelty of the proposed block should be clarified. The proposed blocks still resemble the channel attention, maybe a variant of attention blocks. So we need to differentiate ours with attention and compare with them.
2. The computation cost should be considered by extra blocks in the inference time. How much does burden the proposed blocks bring
3. The training process should be considered, how do we optimize the blocks, how do they converge and why not we do end-to-end training by training two blocks step by step. Is the problem for the weak optimization ability of the blocks?
4. The numerical studies still provide some problems. For the adversarial test, why do we use SVM, what if we adopt a more power classifier,  I think the part should provide more detail that why the experiments are designed as these.
5. The scalability is also a problem, why image-10 with larger improve than image-1K, should be a large dataset or larger model tested?
6. More ablation studies should be done to show the effects of different blocks
7.  Some error bar should be ploted on the figures.
8. The writing should be improved. A more comprehensive review of related works is needed, the training process can be showed by a illustrator.

---

> ### Comment · Action_Editor_rn1a · 2026-01-14
>
> Dear Reviewer yFkT,
>
> Could you please submit your official recommendation (just need a few minutes)? Thank you!
>
> AE

---

### Comment · Action_Editor_rn1a · 2025-10-04

A submission needs 3 reviews and the review ddl is actually Oct 7. I don't know why the rebuttal started before 3 reviews are collected. Perhaps it was a system issue.

---

> ### Comment · Action_Editor_rn1a · 2025-10-04
>
> And reviewers would not discuss further with authors. They would only vote for acceptance/rejection after the rebuttal.

---

### Comment · Action_Editor_rn1a · 2025-10-06

Hi reviewer fCR6,

You have not yet posted your **assignment acknowledgement**, and your review deadline is **within 24 hours**. Please let me know ASAP whether you are able to submit your review by the deadline or not.

AE

---

> ### Comment · Action_Editor_rn1a · 2025-10-20
>
> Hi reviewer fCR6,
>
> Let me know whether you can receive openreview messages ASAP!
>
> AE

---

### Comment · Action_Editor_rn1a · 2025-10-18

Hi reviewer yFkT,

You are **five days** late on your **assignment acknowledgement**. Please complete your task ASAP.

AE

---

> ### Comment · Action_Editor_rn1a · 2025-10-20
>
> Hi reviewer yFkT,
>
> Let me know whether you can receive openreview messages ASAP!
>
> AE

---

> ### Comment · Action_Editor_rn1a · 2025-11-10
>
> Dear reviewer yFkT,
>
> Your review is overdue for 2 days.
>
> Being late is completely understandable and acceptable, but can you first let me know your plan like how late it would be? Otherwise, I may have to find the 6th reviewer for this submission.
>
> Thank you!
>
> AE

---

### Decision · Action_Editor_rn1a · 2026-01-29

**Recommendation:** Reject

**Audience:**

Yes

**Audience Explanation:**

Yes, the reviewers explicitly mentioned that
- The paper really design a feedback mechanism with the FFN, which is different from the current ML architecture, so it sounds like interesting and highly relative.
- People in the filed of Vision backbone design may be interst in this paper.
- While the feedback method itself is not terribly novel, I think the analysis in terms of confidence curves, looking at the sparsity of feature maps chosen, and impact on saliency maps is interesting.

**Claims And Evidence:**

No

**Claims Explanation:**

Claims are not fully supported by empirical evidence and some claims are even misleading. Request changes raised by the reviewers are not fully addressed. Thus, two reviewers have voted for rejection in their official recommendations.

***Claims that are not fully supported***

- **Works for any CNN classifier** This is an extremely broad claim. If experiments are limited to a small set of backbones, the evidence supports “works on these tested architectures”.
- **Improves confidence / calibration** If the submission asserts better confidence or calibration but only shows a small number of plots/metrics, or lacks rigorous calibration reporting, then the claim overreaches.
- **Efficient / low overhead (especially at inference)** If the proposed method requires a second forward pass or extra backbone evaluation, then it is not low overhead in the way most readers interpret it.
- **Detects adversarial noise as a robust security signal** If the submission demonstrates detection using auxiliary classifiers on feedback signals, it does not automatically translate to robustness or security.
- **Low-dimensional approximation of internal representations** This is a representational claim. If the evidence is mostly qualitative, it is suggestive, not conclusive.
- **Novel mechanism** If comparisons are incomplete, the submission may be proposing a new variant rather than a completely new conceptual mechanism.

***Claims that are misleading***

- All the claims mentioned above are potentially misleading, in particular, items 2 to 5.
- Dismissing prior methods as application-specific without evidence. It reads like rhetorical differentiation rather than evidence-based taxonomy. Note that the novelty is based on real scientific contributions and moreover the novelty is just optional for TMLR.

**Resubmission Of Major Revision:**

The authors may consider submitting a major revision at a later time.